# FACT or Fiction: Can Truthful Mechanisms Eliminate Federated Free Riding?

**Marco Bornstein**
University of Maryland
marcob@umd.edu

**Amrit Singh Bedi**
University of Central Florida
amritbedi@ucf.edu

**Abdirisak Mohamed**
University of Maryland
SAP Labs, LLC
amoham70@umd.edu

**Furong Huang**
University of Maryland
furongh@umd.edu

## Abstract

Standard federated learning (FL) approaches are vulnerable to the free-rider dilemma: participating agents can contribute little to nothing yet receive a well-trained aggregated model. While prior mechanisms attempt to solve the free-rider dilemma, none have addressed the issue of truthfulness. In practice, adversarial agents can provide false information to the server in order to cheat its way out of contributing to federated training. In an effort to make free-riding-averse federated mechanisms truthful, and consequently less prone to breaking down in practice, we propose FACT. FACT is the first federated mechanism that: (1) eliminates federated free riding by using a penalty system, (2) ensures agents provide truthful information by creating a competitive environment, and (3) encourages agent participation by offering better performance than training alone. Empirically, FACT avoids free-riding when agents are untruthful, and reduces agent loss by over 4x.

## 1 Introduction

Truth is essential to the functioning of fair systems. In its absence, fraud and corruption lead to negative effects on those left participating. This too is true within machine learning (ML). Adversaries can attack ML algorithms for insidious purposes [3, 7], and algorithms trained on data that is not well-representative or poisoned can lead to dangerous and unfair results [28, 29, 37].

Issues of truthfulness have begun to bleed into multi-agent collaborative learning systems. Federated Learning (FL), the preeminent collaborative learning framework, is susceptible to the free-rider dilemma: agents can contribute little to nothing during training while still obtaining a well-trained final aggregated model [8, 12, 17]. Furthermore, Karimireddy et al. [12] proves that standard FL algorithms naturally yield catastrophic free-riding equilibria, where most rational agents contribute *nothing*. It is important to note that we do not consider the possibility of malicious gradient updates from adversarial agents, covered by recent works [6, 23, 25, 27]. We assume that agents share a common goal of reducing loss, and act selfishly rather than maliciously.

Recent work has aimed to remedy the free-rider dilemma in FL through contract theory [4, 11, 16, 19], collaborative fairness [21, 26, 36], and mechanisms that incentivize contributions [2, 12]. The mechanisms presented in Karimireddy et al. [12] and Bornstein et al. [2] are notable because they do not require **(i)** carefully optimized contracts (agents will always participate because the mechanism is individually rational) or **(ii)** alterations to standard FL training (simple FedAvg [22] can be run with

38th Conference on Neural Information Processing Systems (NeurIPS 2024).

no computation of agent reputations). However, these mechanisms assume that agents are honest and truthful with an orchestrating central server. If agents can lie, the proposed mechanisms no longer guarantee elimination of free riding.

In our paper, we propose a mechanism, **F**ederated **A**gent **C**ost **T**ruthfulness (FACT), that provably eliminates free riding *even when agents are untruthful*. Unlike standard FL approaches, where agents are incentivized to use little to none of their own data, agents participating in FACT are incentivized to use as much data for federated training as they would on their own. Furthermore, when agents use their locally-optimal amount of data, FACT is individually rational (IR): agents always receive greater benefit participating in FACT than by training alone. These results hold even if agents try to cheat the mechanism by misreporting individual training costs to the central server. FACT ensures that each agent's best strategy is to be truthful with the central server.

**Summary of Contributions**. In summary, the main contributions of our paper are,

**(No Free Riding):** Creating a novel penalization scheme that shifts the catastrophic free-riding optimum back to each agent's local optimum (thereby eliminating free riding).

**(Truthfulness):** Constructing a truthfulness mechanism (competition) that dissuades agents from lying about their individual costs with the central server.

**(Realistic):** Showcasing the efficacy of FACT empirically in eliminating agent free riding under untruthfulness, consequently improving federated training performance.

## 2 Related Works

**Distributed Learning Mechanisms for Agent Contribution**. An important line of mechanism research in distributed learning regards mechanisms that incentivize agent contributions for federated training [2, 12, 18, 38, 39]. Some of the first papers in this area were application-based, namely in mobile crowdsensing and smart sensors in vehicles. We detail these works in Appendix A. Zhan et al. [39] is one of the first works to define agent and server utility when data is shared between agents and the server to train a global model (including agent training costs). The server announces a total reward, and agents determine how much data to contribute in order to maximize their reward. A deep reinforcement learning method is used by the server to learn the optimal announced reward.

An accuracy-shaping mechanism is proposed in Karimireddy et al. [12], which also follows a similar data sharing and iid setting. This mechanism does not require the use of contracts or learning of a reward, and guarantees that an agent will receive improved reward when participating in the mechanism than by not participating. In the accuracy-shaping mechanism, the server degrades model performance for an agent if it does not contribute as much data as locally optimal. Bornstein et al. [2] proposes a similar accuracy-shaping mechanism. However, unlike Zhan et al. [39] and Karimireddy et al. [12], agent utility is modeled in a non-linear and more realistic manner. Additionally, the accuracy-shaping mechanism is generalized to the federated setting (no data sharing) where data can be non-iid. Unlike the majority of the works above, our mechanism seeks not just to incentivize agent contribution, but to eliminate free riders. Moreover, unlike [2, 12], we do not assume that agents are truthful. Our mechanism ensures that agents contribute what is locally optimal, thereby eliminating free-riding, and punishes agents for lying to the server about their costs.

**Truthful Distributed Learning Mechanisms**. Similar to the agent contribution mechanisms above, many existing truthful distributed learning mechanisms exist within the crowdsensing setting [9, 33, 35, 42]. We detail these works in Appendix A. The more recent works [14, 20, 34] examine truthful mechanisms within traditional FL. Lu et al. [20] explores truthful mechanisms in Vertical FL, where features are split up amongst agents. Specifically, a truthful mechanism is designed (via a novel transfer-payment rule) to maximize social utility, without use of an auction, even when agents can lie about their costs. Within Horizontal FL (the standard approach), Le et al. [14] designs a randomized auction for resource-constrained wireless agents, which achieves approximate-truthfulness in expectation. The server receives bids from agents to perform training, and selects a winning bid (agent) for each uplink sub-channel. The server pays a reward designed to minimize agent social cost, and does not take into account the server's utility. Wu et al. [34] builds on the auction idea in Le et al. [14], instead using a federated auction bandit which learns to select bidding agents

that maximize the server's utility. The mechanism ensures truthfulness and individual rationality. Like the works above, our mechanism ensures truthfulness and individual rationality of participating agents. In contrast, our mechanism setting is within horizontal FL and does not require an auction or bandit method to ensure truthfulness. As such, any willing agent can participate in training (no bids are rejected), an optimal server reward does not need to be learned, and agents do not need to have a private valuation. Finally, our mechanism requires no direct payment from the server (all reward payments come from other agents) and provably eliminates the free-riding dilemma. Due to space constraints, **we detail works regarding fairness and agent selection mechanisms in Appendix A.**

## 3   The Challenge of Free Riding in Federated Learning

**Federated Setting**. Our work explores the setting of Centralized Federated Learning (CFL), where $n$ agents collectively train a model $w$ under orchestration of a central server. Within each iteration $t$ of CFL, the server sends down the current aggregated model $w^t$ to each agent. Agents then perform $h$ local gradient updates (with their own local data) to generate an updated model $w_i^{t+1} \ \forall i \in [n]$. Models are aggregated by the server and this process is repeated until convergence. Each agent is assumed to have access to the same data distribution $\mathcal{D}$, and thus we assume data is independent and identically distributed (iid). We leave construction of truthful mechanisms in the non-iid setting for future research. Our work aims to bring truthful mechanisms to CFL, as none have existed in either the iid or non-iid settings prior to this work. We now begin by detailing the objective that agents participating in FL seek to minimize.

**Federated Objective**. The goal of FL is to train a global model, leveraging the data and compute power of thousands of agents, that achieves low loss. To start, we utilize the convergence of distributed stochastic gradient descent (D-SGD) [15] to first-order stationary points in non-convex settings,

$$\frac{1}{T} \sum_{t=0}^{T-1} \mathbb{E} \|\nabla f(w^t)\|^2 \leq \frac{2\Delta_f}{\gamma T} + \frac{\gamma \sigma^2 L}{2 \sum_{j=1}^n m_j}. \tag{1}$$

The parameter $\Delta_f$ is the difference between the optimal objective value $f^*$ and starting value $f(w^0)$. The global objective function $f$ is the uniform sum over all device objective functions $f_i$. There are $T$ total iterations. The step-size is $\gamma$ and $L$ is the Lipschitz constant, where $\gamma L < 2$. The effective gradient variance is $\sigma^2 / \sum_{j=1}^n m_j$. This stems from bounded variance assumptions, formally defined as $\mathbb{E} \|\frac{1}{M} \sum_{i=1}^M \nabla f(w; \xi_i) - \nabla f(w)\| \leq \frac{\sigma^2}{M}$ (for $M$ batches of data $\xi$).

**Local Agent Loss**. All rational agents seek to minimize their loss. As shown in Equation (1), this can be accomplished by using more data, and thus a larger batch size $m$, as well as more rounds of gradient updates $T$. However, all agents incur a cost $c_i$ for collection, sampling, and compute costs for each data sample $m$ used. The linear cost $c_i m$ stems from our cross-agent setting. Within this setting, IoT devices are prevalent and data is not difficult to collect (*e.g.,* IoT sensors). Thus, it makes the most sense that, on average, there are constant costs over time to collect and sample data (*e.g.,* the cost to power a sensor is usually constant on average) [30, 20, 34].

Overall, agents must balance the benefit of using more data to achieve better model performance with the costs of data collection, sampling, and computations. We formulate this tradeoff for each agent by defining a local loss function $\ell_{i,l}(m_i)$, that depends on the data collected $m_i$ by each agent $i$. Each agent minimizes its loss function, which is a combination of data costs and convergence error terms. When training locally, or alone, an agent loses the benefit of an increased batch size across all agents $\sum_{j=1}^n m_j$. Thus, after removing constants, Equation (1) transforms into our local agent loss,

$$\ell_{i,l}(m_i) := \frac{\gamma \sigma^2 L}{2m_i} + c_i m_i. \tag{2}$$

The optimal amount of data each agent uses for local training can be determined with calculus.

**Theorem 1** (Optimal Local Data Usage). *For an agent $i$ with marginal cost $c_i$, the optimal amount of data $m_{i,l}^*$ used for local training is $m_{i,l}^* := \sqrt{\frac{\gamma\sigma^2 L}{2c_i}}$.*

**Remark 1.** *Agents' optimal amount of local data is inversely proportional to its cost $c_i$. Larger costs lead to less data used and vice versa. If costs were zero, agents would use infinite data.*

**The Free-Rider Dilemma: Shifted Optima in Federated Settings**. When agents engage in federated training, the effective batch size returns to the sum over all agent data $\sum_{j=1}^n m_j$. By participating in federated training, agents gain the benefit of an increased batch size. As such, each agent $i$'s loss function in the federated setting becomes,

$$\ell_{i,F}(m_i) := \frac{\gamma\sigma^2 L}{2(m_i + \sum \boldsymbol{m}_{-i})} + c_i m_i. \tag{3}$$

While slight, this transformation changes the optimal amount of data used by each agent.

**Theorem 2** (Free-Riding: Optimal Federated Data Usage). *For an agent $i$ with marginal cost $c_i$, the optimal amount of data $m_{i,F}^*$ used for federated training is $m_{i,F}^* = \sqrt{\frac{\gamma\sigma^2 L}{2c_i}} - \sum \boldsymbol{m}_{-i}$.*

**Remark 2.** *By participating in federated training, each agent is incentivized to use fewer data samples. This explicitly demonstrates the free-rider dilemma mathematically: it may be optimal for an agent to use no data $m_{i,F}^* = 0$ (if $\sum \boldsymbol{m}_{-i} \geq \sqrt{\frac{\gamma\sigma^2 L}{2c_i}}$) and still possibly achieve lower cost overall (if $\sum \boldsymbol{m}_{-i} \geq m_{i,l}^*$).*

Due to space constraints, we leave the proofs of Theorems 1 and 2 in Appendix C.

## 4 Eliminating Free Riding via Penalization

The natural equilibrium of traditional FL is free riding: agents use less data than is locally optimal in proportion to how much data other agents use (Theorem 2). This necessitates the construction of a mechanism to restore the equilibrium of FL back to what is locally optimal for each agent. In this section, we detail how penalization can perform this restoration.

**Penalized Federated Learning (PFL)**. To disincentivize agents from straying from their locally optimal data usage $m_{i,l}^*$, the central server can use contract theory to ensure each agent $i$ uses its locally optimal amount for a reported cost $c_i$. Namely, when agreeing to participate in FL, each agent $i$ agrees to pay the following free-riding penalty $P_{fr}$ if they do not use $m_{i,l}^* = \sqrt{\frac{\gamma\sigma^2 L}{2c_i}}$ data samples,

$$P_{fr}(m_i) := \lambda_i \left( \frac{c_i}{2\lambda_i} - \frac{\gamma\sigma^2 L}{4\lambda_i(m_{i,l}^* + \sum \boldsymbol{m}_{-i})^2} + m_{i,l}^* - m_i \right)^2. \tag{4}$$

The hyperparameter $\lambda_i > 0$, chosen by the central server, denotes the harshness of the free-riding penalty. The value of $\lambda_i$ is known by each agent $i$ while deciding whether or not to participate in the mechanism. Since the penalty is levied on each agent by the server, it is incorporated into their federated loss function. This results in a new amended federated loss function $\ell_{i,PFL}(m_i)$ defined as,

$$\ell_{i,PFL}(m_i) = \underbrace{\frac{\gamma\sigma^2 L}{2(m_i + \sum \boldsymbol{m}_{-i})} + c_i m_i}_{\ell_{i,F}(m_i)} + P_{fr}(m_i). \tag{5}$$

**Theorem 3** (PFL Eliminates Free Riding). *For an agent $i$ with marginal cost $c_i$, the optimal amount of data $m_{i,PFL}^*$ used for federated training under Equation (5) is $m_{i,PFL}^* := \sqrt{\frac{\gamma\sigma^2 L}{2c_i}}$.*

---
**Algorithm 1** PFL: Penalized Federated Learning
---
1: **Input:** data $\boldsymbol{m} = [m_1, \ldots, m_n]$, marginal costs $\boldsymbol{c} = [c_1, \ldots, c_n]$, $h$ local steps, $T$ iterations, $E$ epochs, initial parameters $\boldsymbol{w}^1$, loss $f$, step-size $\gamma$, and constants $(L, \sigma^2, \alpha)$.
2: **Output:** model $\boldsymbol{w}^T$.
3: Server determines expected optimal data use for each agent $m_i^* \leftarrow \sqrt{\frac{\gamma \sigma^2 L}{2c_i}}$
4: Server computes each agent's penalty scalar $\lambda_i \leftarrow \frac{m_i^*(\sum \boldsymbol{m})}{(2-\alpha)\gamma\sigma^2 L \sum \boldsymbol{m}_{-i}} \left( c_i - \frac{\gamma\sigma^2 L}{2(\sum \boldsymbol{m}_{-i} + m_i^*)^2} \right)^2$
5: Server receives free-rider penalty payment $P_{fr}(m_i)$ from each agent via Equation (4) using $\lambda_i$
6: $s_i \leftarrow m_i / \sum_{j=1}^n m_j$
7: **for** $t = 1, \ldots, T$ **do**
8:     Server distributes $\boldsymbol{w}^t$ to all agents
9:     **for** $h$ local steps, each agent $i$ **in parallel do**
10:         $\boldsymbol{w}_i^{t+1} \leftarrow$ Agent Update$(\boldsymbol{w}^t, i, m_i, f, h)$          {AgentUpdate is detailed in Algorithm 3}
11:     **end for**
12:     $\boldsymbol{w}^{t+1} \leftarrow \sum_i s_i \boldsymbol{w}_i^{t+1}$
13: **end for**
---

> **Remark 3.** *By utilizing the PFL loss in Equation (4), agents participating in FL will use a locally optimal data amount ($m_{i,PFL}^* = m_{i,l}^*$). This eliminates the free-rider dilemma seen in Theorem 2.*

Agents participating in PFL (Algorithm 1) follow the amended loss shown in Equation (5). Careful selection of $\lambda_i$ is required to ensure that PFL is IR.

**Ensuring Individual Rationality (IR) in PFL.** As shown above in Theorem 3, it is optimal for agents participating in the penalized federated scheme (with a loss function shown in Equation (5)) to contribute what is locally optimal *i.e.*, $m_{i,PFL}^* = m_{i,l}^*$. To ensure individual rationality, it must be true that participating in federated training will produce at least as much reward for an agent than by not participating. The selection of $\lambda_i$ is crucial in accomplishing this task.

> **Lemma 1** (PFL Assurance of IR at Optimum). *Let $c_i$ be the marginal cost for an agent $i$, $m^* := \sqrt{\frac{\gamma\sigma^2 L}{2c_i}} = m_{i,l}^* = m_{i,PFL}^*$ be the agent's locally optimal data usage, and $\alpha \in [0, 2)$ be a server-specified hyper-parameter. In order to ensure that the optimal penalized federated loss is at least lower than the optimal local loss, $\ell_{i,PFL}(m_i^*) \leq \ell_{i,l}(m_i^*)$, one must select $\lambda_i$ such that,*
>
> $$\lambda_i := \frac{m^*(\sum \boldsymbol{m})}{(2-\alpha)\gamma\sigma^2 L \sum \boldsymbol{m}_{-i}} \left( c_i - \frac{\gamma\sigma^2 L}{2(\sum \boldsymbol{m}_{-i} + m^*)^2} \right)^2. \tag{6}$$
>
> *Selection of such $\lambda_i$ results in a loss gap between $\ell_{i,PFL}(m^*)$ and $\ell_{i,l}(m^*)$ of,*
>
> $$\Delta \ell_i := \ell_{i,l}(m^*) - \ell_{i,PFL}(m^*) = \frac{\alpha}{4} \left( \frac{\gamma\sigma^2 L \sum \boldsymbol{m}_{-i}}{m^*(\sum \boldsymbol{m}_{-i} + m^*)} \right). \tag{7}$$

> **Remark 4.** *Agents provably receive improved loss $\Delta\ell$ when they use their locally optimal data amount $m_{i,l}^*$ participating in PFL (Algorithm 1) versus local training, thereby ensuring IR.*

> **Remark 5.** *The parameters $\lambda_i$ and $\alpha$ control the amount of benefit received by an agent $i$. A larger value of $\alpha$, and consequently $\lambda_i$, results in a larger gap between $\ell_{i,PFL}(m^*)$ and $\ell_{i,l}(m^*)$. Inversely, a smaller value of $\alpha$ and $\lambda_i$ results in a smaller gap between local and PFL loss.*

Due to space constraints, we leave the proofs of Theorem 3 and Lemma 1 in Appendix C. Now, we formally prove that PFL both (**i**) eliminates agent free riding and (**ii**) is individually rational.

> **Theorem 4** (Elimination of Federated Free-Riding With Truthful Agents). *PFL (Algorithm 1) using $\lambda_i$ from Lemma 1 is IR and eliminates the free-rider dilemma when agents are truthful.*

**Proof.** The result of Theorem 3 is that each agent $i$'s optimal strategy within the penalized federated scheme is to use their locally optimal amount of data $m_i^* = \sqrt{\frac{\gamma\sigma^2 L}{2c_i}}$. Furthermore, Lemma 1 states that using $m_i^*$ within the penalized federated scheme results in a reward, or improvement over local

---

**Algorithm 2** FACT: Federated Agent Cost Truthfulness

---
1: **Input:** data $\boldsymbol{m} = [m_1, \ldots, m_n]$, marginal costs $\boldsymbol{c} = [c_1, \ldots, c_n]$, $h$ local steps, $T$ iterations, $E$ epochs, initial parameters $\boldsymbol{w}^1$, loss $f$, step-size $\gamma$, and constants $(L, \sigma^2, \alpha)$.
2: **Output:** model $\boldsymbol{w}^T$ and monetary reward $r_i$.
3: Server receives contract payment from each agent: $\Delta \ell_i \leftarrow \frac{\alpha}{4} \left( \frac{\gamma \sigma^2 L \sum \boldsymbol{m}_{-i}}{m_i (\sum \boldsymbol{m})} \right)$
4: Run Penalized Federated Learning: $\boldsymbol{w}^T \leftarrow \text{PFL}(\boldsymbol{m}, \boldsymbol{c}, h, T, E, \boldsymbol{w}^1, f, \gamma, L, \sigma^2, \alpha)$
5: Compute each agent reward $r_i$ via the cost truthfulness game in Equation (10)

---

training, of $\Delta \ell_i = \frac{\alpha}{4} \left( \frac{\gamma \sigma^2 L \sum \boldsymbol{m}_{-i}}{m_i^* (m_i^* + \sum \boldsymbol{m}_{-i})} \right)$. Thus, by combining Theorem 3 and Lemma 1, truthful agents which choose to participate in PFL attain a reshaped and lower loss at the same optimum. Individually rational agents would therefore prefer to participate in PFL (Algorithm 1) over local training due to the reshaped optimum's lower loss. This optimum achieves the same data usage as local training, thereby eliminating the free-rider dilemma.

> **Remark 6** (Truthfulness Concerns). *Participating in PFL, rational agents contribute as much as they would locally (Theorem 3) and attain more benefit (Lemma 1). The only issue that remains is truthfulness: agents may lie about their costs in order to contribute less to federated training under the guise of what looks like an "optimal amount" to the server (thereby avoiding large $P_{fr}$).*

While PFL eliminates free riding when agents are truthful, this assumption is not realistic in practice. Untruthful agents can report inflated costs to the server in order to trick it into expecting smaller data usage. In effect, agents can lie about their costs to free ride. In the next section, we propose a mechanism which incentivizes agents to be truthful (*i.e.,* their best strategy is not to lie).

# 5   FACT: Eliminating Free-Riding With Untruthful Agents

When agents are truthful, PFL (Algorithm 1) provably eliminates the free-rider dilemma (Theorem 4). However, an adversarial agent $i$ can still free ride by misreporting, or lying, about their true marginal cost $c_i$. As previously mentioned, we do not consider the situation of malicious gradient updates coming from adversarial agents, covered by recent works [6, 23, 25, 27]. We assume that agents share a common goal of reducing loss, and act selfishly rather than maliciously.

As seen in Theorem 2, an adversarial agent can lie their way to a smaller data optimum by inflating their cost: $m_i^{lie} = \sqrt{\frac{\gamma \sigma^2 L}{2(c_i + \epsilon)}} < m_{i,l}^*$. In this case, an adversarial agent will **(i)** avoid server penalty (since it will be contributing what looks like an optimum from the server's view), **(ii)** incur smaller true costs $c_i m_i^{lie} < c_i m_i$, and **(iii)** still reap the benefits of a larger batch size $\sum \boldsymbol{m}_{-i} + m_i^{lie}$.

> **Definition 1** (Truthful Mechanism). *A truthful mechanism is one in which no agent can reduce its loss by reporting a cost different from its true cost, regardless of other agents' actions. Overall, each agent's dominant strategy is to report its true cost.*

To counteract the possibility of an agent being untruthful with the server about its true cost, we propose a truthful mechanism FACT: Federated Agent Cost Truthfulness (Algorithm 2). FACT ensures that each agent $i$'s dominant strategy is to report its true cost $c_i$ (thereby satisfying Definition 1).

> **Assumption 1** (No Collusion). *Agents have no knowledge of other agents' costs or the distribution of agent costs $f_C$. In the absence of information, each agent $i$ believes that its cost $c_i$ is equally likely to be larger or smaller than any other agent's cost.*

Our lone assumption about agent knowledge is that agents are unknowledgeable about the distribution of agent costs $f_C$ or any other agent's cost: agent costs are private. Assumption 1 ensures that there is no collusion amongst agents. In the absence of knowledge about other agents' costs, it is reasonable for an agent to believe that their private cost is equally likely to be greater or lesser than any other agent's cost (*i.e.,* the median cost). Now that agents can report costs $c$ which differ from their true cost

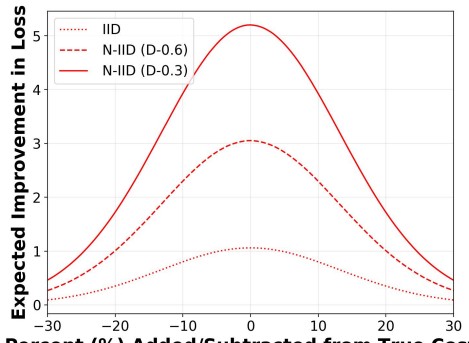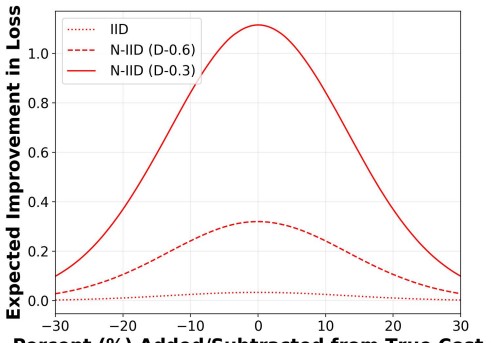

Figure 1: **Enforcement of Agent Truthfulness.** Average net improvement in loss over local training is plotted for iid (dotted line) and two non-iid agent distributions (D-0.6: dashed, D-0.3: solid). For both CIFAR10 (left) and MNIST (right), agents maximize their net improvement in loss when they are truthful (0% added) about their true cost. This matches our theory in Theorem 5.

$c_i$, each agent's PFL penalty (Equation 4) and loss (Equation 5) becomes a multivariable function,

$$P_{fr}(m_i, c) = \lambda_i \left( \frac{c}{2\lambda_i} - \frac{\gamma \sigma^2 L}{4\lambda_i (\sqrt{\frac{\gamma \sigma^2 L}{2c}} + \sum \boldsymbol{m}_{-i})^2} + \sqrt{\frac{\gamma \sigma^2 L}{2c}} - m_i \right)^2, \tag{8}$$

$$\ell_{i,PFL}(m_i, c) = \frac{\gamma \sigma^2 L}{2(m_i + \sum \boldsymbol{m}_{-i})} + c_i m_i + P_{fr}(m_i, c). \tag{9}$$

Regardless of the reported cost $c$, each agent $i$ will incur a true cost $c_i m_i$ locally in Equation (9).

**Using a Sandwich to Incentivize Truthfulness**. Our mechanism is the first to ensure truthfulness while solving the free-rider dilemma in FL. FACT ensures agent truthfulness by fostering a competition amongst agents. The competition begins at the end of federated training (PFL) by randomly grouping reported agent costs into threes. The rules of this competition, or truthfulness mechanism, are simple: each agent $i$ "wins" the competition, and receives a reward, if its cost $c_i$ is sandwiched between the two other agent costs in its group. If there is a tie, the server randomly selects one of the tied agents as the winner. This mechanism is detailed mathematically below,

$$P_{ct}(c) := \begin{cases} \Delta \ell_i - \frac{3}{n} \sum_{j \neq i \in [n]} \Delta \ell_j & \text{if } C_a < c < C_b, \\ \Delta \ell_i & \text{else.} \end{cases} \quad C_a, C_b \sim f_C \text{ randomly sampled costs} \tag{10}$$

**Lemma 2.** *The probability of an agent "winning" in the truthfulness mechanism in Equation (10), given a reported cost $c$, is $\upsilon(c) := 2F_C(c)(1 - F_C(c))$, where $F_C$ is the CDF of $f_C$.*

The crux of the truthfulness competition is that each agent $i$ *never receives its own reward $\Delta \ell_i$*. Instead, as shown in Equation (10), each agent pays $\Delta \ell_i$ to the server before training regardless of the mechanism outcome. After training, if the agent "wins" the competition, by having a cost sandwiched between $C_a$ and $C_b$, then it receives triple the average of all other agent rewards $\frac{3}{n} \sum_{j \neq i \in [n]} \Delta \ell_j$. The average reward is tripled, since two other agents will lose the competition and receive no reward. By making the reward decoupled from an agent's own reward $\Delta \ell_i$, each agent can no longer increase or decrease its reward by altering how much data $m_i$ they use for federated training. Now, they can only affect the likelihood of winning the competition by choosing what cost $c_i$ to report to the server.

Incorporating the truthfulness mechanism into the PFL loss function results in the FACT loss function,

$$\ell_{i,Fact}(m_i, c) = \underbrace{\frac{\gamma \sigma^2 L}{2(m_i + \sum \boldsymbol{m}_{-i})} + c_i m_i + P_{fr}(m_i, c)}_{\ell_{i,PFL}(m_i,c)} + P_{ct}(c). \tag{11}$$

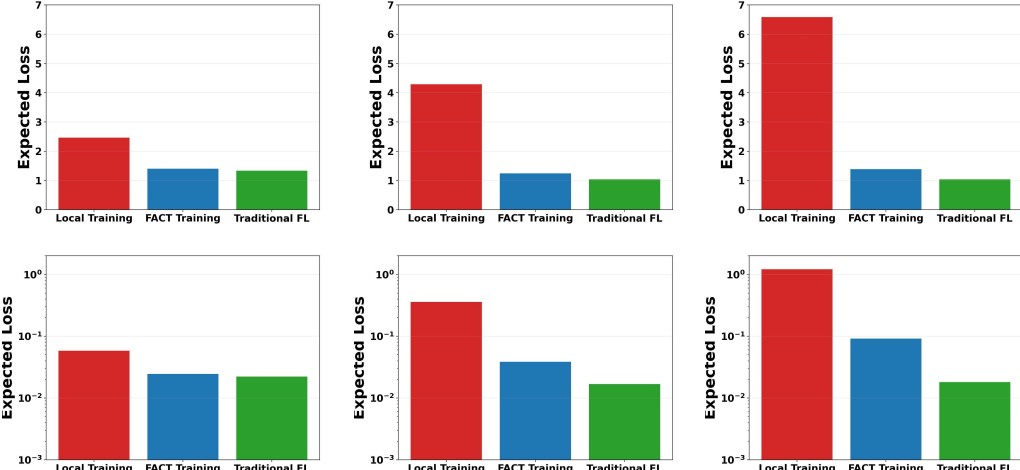

Figure 2: **Reduction in Agent Loss.** The average agent loss for baselines on CIFAR10 (top row) and MNIST (bottom row) under iid (left), and two non-iid data distributions (center: D-0.6, right: D-0.3). Traditional FL is an upper bound on agent loss (if agents did not free ride). FACT improves agent loss over local training by up to a factor of 3x for CIFAR10 and 4x for MNIST.

**Theorem 5** (Main Theorem). *Each agent $i$'s best strategy, when participating in* FACT *(Equation 11), is to report its true cost and use its locally optimal amount of data* $(m_i, c)_i^* = (\sqrt{\frac{\gamma\sigma^2 L}{2c_i}}, c_i)$.

**Remark 7** (Truthfulness). FACT *is a truthful mechanism: each agent's best strategy is to report its true cost $c_i$. Agents cannot reduce their loss by reporting a different cost (satisfying Definition 1).*

**Remark 8** (Elimination of Free Riding). FACT *eliminates agent free riding: each agent's best strategy is to use as much data as is locally optimal $m_i^* = \sqrt{\frac{\gamma\sigma^2 L}{2c_i}}$ (Theorem 1). Thus, the improved optimum for PFL (Theorem 3) is maintained even under agent untruthfulness.*

**Remark 9** (Individually Rational). *When using their best strategy, agents participating in* FACT *either (i) receive loss equivalent to local training if they lose the competition or (ii) reduced loss if they win (shown in Equation 10). Thus,* FACT *is IR as agents using their best strategy never receive worse loss than if they did not participate. In fact, agents receive lower loss in expectation.*

## 6 Experimental Results

Below, we analyze the effectiveness of FACT at **(1)** enforcing agent truthfulness, **(2)** eliminating free riding, and **(3)** reducing agent loss compared to local training. We perform true federated experiments (no simulations) using CIFAR10 [13] and MNIST [5] datasets to train an image classification model.

**Experimental Setup**. Within our experiments, 16 agents train a model individually (locally) as well as in a federated manner. Each agent uses 3,125 and 3,750 data samples each for CIFAR10 and MNIST respectively. We analyze FACT under homogeneous and heterogeneous agent data distributions. For heterogeneous agent data distributions, we use Dirichlet distributions with parameters $\alpha = 0.3, 0.6$ to determine the label ratio for each agent [10]. We simulate the sandwich truthfulness competition defined in Equation (10) by randomly sampling 2,000 synthetic agent costs from a Gaussian distribution with a mean centered at each agent's true cost (standard deviation of one-tenth of the true cost). We perform 100,000 simulation trials and compute mean performance for each agent over all trials. Further experiment details and hyperparameters are found in Appendix B.

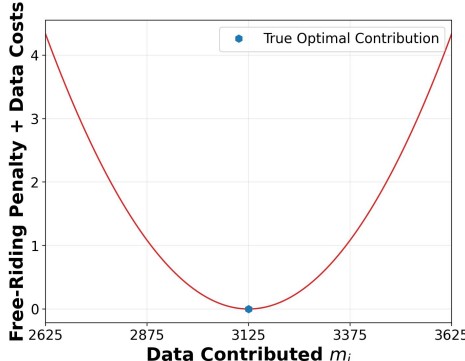
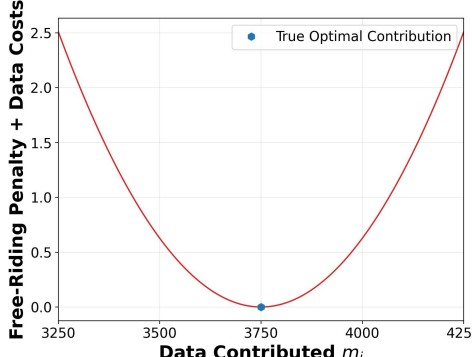

Figure 3: **Elimination of Free Riding via Penalty.** The penalty term $P_{fr}(m_i)$ plus data collection costs $c_i m_i$ is plotted for CIFAR10 (left) and MNIST (right) for varying data contributions $m_i$. These combined costs are minimized at the local optimum $m_{i,l}^*$, as predicted by Theorem 3.

**Lack of Baselines**. FACT is the first truthful Horizontal FL method that eliminates free riding while simultaneously enforcing truthfulness without the use of an auction or bandit (thereby allowing all willing agents to participate). As such, we compare the results of FACT with the only available option: local training. We showcase below that FACT reduces agent loss compared to local training while simultaneously avoiding free-riding and truthfulness issues.

**FACT Enforces Truthfulness**. The results of Figure 3 back the theoretical result of Theorem 5, which states that agents maximize their expected reward (*i.e.,* net reduction in loss versus local training) when truthfully reporting their cost. Across both homogeneous (iid) and heterogeneous (n-iid) agent data distributions, agents' net improvement in loss peaks when reporting their true cost (0% added) and monotonically decreases as agents report an inflated or deflated true cost. Thus, the sandwich-style truthfulness competition FACT employs disincentivizes agents from inflating or deflating their true costs. This is important, as agents are no longer incentivized to lie to the server about their true cost in order to free-ride without the free-rider penalty in Equation (4) being imposed. Net improvement in loss grows as data distributions become increasingly non-iid. FL is robust enough to deal with the increasingly non-iid datasets while local training suffers (Figures 5 & 6). Therefore, the gap between FL and local training grows, and thus so too does the reward in FACT.

**FACT Reduces Agent Loss**. Agents participating in FACT on average achieve a reduction in total loss compared to training on their own when using the same amount of data. Consequently, participating in FACT is individually rational: agents at worst receive loss equivalent to local training and on average receive improved loss (Figure 2). Traditional FL achieves a lower average agent loss than FACT when *all agents use as much data as they would locally*. However, as shown in Theorem 2, this is unrealistic. Traditional FL emits a catastrophic free-riding optimum. The results of traditional FL in Figure 2 is the upper bound on achievable loss by agents, only attainable if agents refused to free ride. The gap in loss between FACT and the upper bound of traditional FL is the cost of agent untruthfulness and their inclination to free ride. The gap is the direct result of the penalty terms on free riding $P_{fr}$ and truthfulness $P_{ct}$ found in Equation (11). Average local loss increases as data distributions become increasingly non-iid, while traditional FL remains robust to the distribution shift (Figures 5 & 6). Since the agents in FACT who lose the truthfulness competition receive a loss equivalent to local training, the loss of FACT also rises in proportion to the local training loss.

**FACT Eliminates Free Riding**. Within our experiments for CIFAR10 and MNIST, a marginal cost was carefully selected such that it is locally optimal for each agent to use 3,125 and 3,750 data samples for training respectively. As shown in Figure 3, the free-rider penalty $P_{fr}$ harshly penalizes agents for deviating from this locally-optimal amount. This result accounts for the reduction (gain) in data collection costs for using less (more) data during training. Thus, as proven theoretically in Theorem 3, we confirm that it is suboptimal for an agent $i$ to use either more or less data than is what is locally optimal (for a given reported cost $c_i$) when participating in FACT.

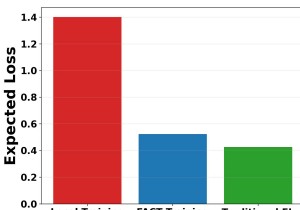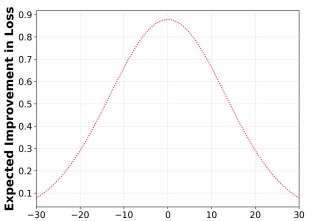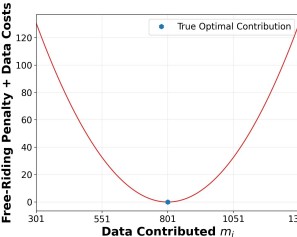

Figure 4: **FACT Eliminates Free Riding in Realistic Settings**. When training an image classifier for diagnosing skin cancer, agents participating in FACT achieve much lower loss ($66\%$ less) than if they did not participate (left). Agents maximize their improvement in loss over local training when they are truthful; reporting inflated or deflated costs diminishes improvement in loss (middle). Agents minimize penalties when using their locally optimal amount of data ($m^* = 801$) for training (right).

## 6.1 Real-World Case Study: Inter-Hospital Skin Cancer Diagnosis

We consider a realistic situation where a consortium of hospitals seek collaboration to train a model, privately in a FL manner, that can diagnose skin cancer. One of the hospitals is smaller and resource-constrained. It is difficult, but not impossible, for this hospital to collect more data for training. Thus, in the absence of a truthful FL mechanism, the smaller hospital could over-report its collection costs to the server in order to contribute little to no data towards training while still reaping the rewards of a well-trained global model.

To test how FACT deals with this scenario, we train an image classification model on the HAM10000 [31] dataset. HAM10000 consists of a collection of dermatoscopic images of pigmented lesions. The goal is to train a model which can perform automated diagnosis of pigmented skin lesions, including melanoma (skin cancer). Our setup is similar to the experimental setup detailed in the previous section. One difference is that we fine-tune a ResNet50 model on HAM10000 that is pre-trained on ImageNet (a realistic approach for a hospital). HAM10000 is an imbalanced dataset, and evenly partitioning $80\%$ of the data amongst 10 devices, as their local training sets, further exacerbates the non-iidness of the data. This too is realistic, as data is often non-iid amongst FL agents. We use the Adam optimizer with a learning rate of 1e-3 and batch size of 128.

As expected, FACT provably dissuades any hospital from lying about its cost in order to reduce their data contribution level. Agents are incentivized to be truthful; reduction in loss is maximized when agents truthfully report their costs (see the middle plot in Figure 4). As a result, a smaller, resourced-constrained hospital would still contribute its locally optimal amount of data, or face harsh penalties (see the right plot in Figure 4). Finally, all participating hospitals are still better off participating in FACT than training alone. Participating agent loss is reduced by nearly $66\%$ compared to local training (see the left plot in Figure 4). The effectiveness of FACT to dissuade free-riding in the real-world setting of training a skin cancer classifier across hospitals indicates that it will be successful in other real-world settings.

## 7 Conclusion

Traditional FL frameworks are susceptible to free riding: it is often an agent's best strategy to contribute little to nothing during training while still obtaining a well-trained final global model. Furthermore, many FL frameworks do not account for untruthful agents which report inaccurate information to a server in order to skip penalties and collect greater rewards. Our proposed mechanism, FACT, is simultaneously individually rational, truthful, and eliminates free riding. FACT leverages a novel penalization and sandwich mechanism to shift each agent's best strategy to report its true cost with the server (**truthfulness**) and use as much data as it would on its own (**no free riding**). Furthermore, agents which participate in FACT will never receive worse loss than by not participating, and receive lower loss on average if they participate (**individually rational**). Empirically, we find that FACT enforces agent truthfulness while reducing agent loss by upwards of a factor of four.

## Acknowledgments

Bornstein and Huang are supported by National Science Foundation NSF-IIS-2147276 FAI, DOD-ONR-Office of Naval Research under award number N00014-22-1-2335, DOD-AFOSR-Air Force Office of Scientific Research under award number FA9550-23-1-0048, DOD-DARPA-Defense Advanced Research Projects Agency Guaranteeing AI Robustness against Deception (GARD) HR00112020007, Adobe, Capital One and JP Morgan faculty fellowships.

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

# Appendix

## A    Additional Related Works

While small, the research area of incentives and mechanism design in FL is growing. There are a few surveys [32, 40, 41] which overview FL mechanisms, current research directions & challenges, and future opportunities. We hone in on a few of these research directions, namely agent contribution and truthful FL mechanisms, as well as fairness and agent selection mechanisms. Due to space constraints, we continue our related works (Section 2) here below.

**Distributed Learning Mechanisms for Agent Contribution (Continued)**. Within Zhan et al. [38], an iterative Nash bargaining solution is proposed where mobile devices bargain with a server over the amount of data to contribute and the payments it should receive. The server dynamically alters the payment price to adjust its gap between desired supply of data and the given demand. Liu et al. [18] designs a reverse-auction approach, with subsidized or "endowment" compensation, that incentivizes non-participating vehicles to reduce roadside server congestion by offloading its data. Both of these works deal with data exchange and do not consider the FL setting, where models are collaboratively trained together. Furthermore, it is not guaranteed in either that agents and the server will both always achieve higher payoff for participating than not (in Liu et al. [18] the server attains smaller payoff by participating).

**Truthful Distributed Learning Mechanisms (Continued)**. Xu et al. [35] deals with agents submitted untruthful bids to a server, specifically where an agent can lie about sensing time windows and costs. The proposed mechanism leverages a Vickrey-Clarke-Groves auction, thereby ensuring truthfulness in bid price. Time-window truthfulness is ensured via trusted time stamping. Similarly, the remaining crowdsensing works leverage auctions to ensure agent bid truthfulness. Zhang et al. [42] introduces a double auction mechanism, MobiAuc, for proximity-based mobile crowdsensing. MobiAuc addresses issues of dynamic mobility patterns (device matching) while maintaining agent truthfulness surrounding their bid (*e.g.,* service valuation or arrival time). In Wang et al. [33], a distributed truthful mechanism is proposed, where agents send bids for their services (usually data) to buyers. Gao et al. [9] designs a reverse auction mechanism that ensures vehicles report their true costs when crowdsensed. The works above do not consider the FL setting, where models are trained through the help of many agents. Thus, these works do not solve the free-rider dilemma, which only exists in FL settings. Furthermore, our mechanism is able to ensure agent truthfulness without the use of an auction.

**Fairness FL Mechanisms**. Recent work within FL has focused on agent fairness [1, 21, 24]. These works seek to fairly allocate model performance to agents depending upon how much they contribute during federated training. Lyu et al. [21] computes a "reputation" metric for each agent (measures the amount of agent contribution). A better reputation leads to improved model performance. Both Blum et al. [1] and Murhekar et al. [24] seek to find agent contribution equilibria that improve the welfare of all agents. Within [1], the existence and stability of equilibria which avoid free riding and agent envy. In [24], a mechanism is proposed to alter agent strategies in order to maximize the net utility of all participating agents. Unlike the works above, FACT is a truthful mechanism.

**Mechanisms for FL Agent Selection**. There exists recent literature which aims to incentivize high-quality agents to participate in federated training [11, 19, 43]. Agents can be deemed as high quality if, for example, they have valuable data or are very willing to participate in training. In Liu et al. [19], a two-dimensional contract model is created to consider agent willingness and data quality when designing contract fees. Kang et al. [11] introduces a scheme where agent reputations are calculated (from previous interactions or other server interactions) and leveraged to select reliable agents for federated training. Zhou et al. [43] designs an auction method that incentivizes a diverse set of agents to participate in training. Truthfulness and individual rationality are ensured by properties of the auction that is constructed. FACT does not select agents based on a given quality or willingness criteria. All agents are able to participate if they choose. Instead, FACT solves agent free riding when agents are truthful as well as untruthful.

---

**Algorithm 3** AgentUpdate

---
1: **Input:** model parameters $w$, agent index $i$, data used for training $m_i$, loss $f$, $h$ local steps.
2: **Output:** updated model parameters $w$.
3: $\mathcal{B} \leftarrow$ batch $m_i$ data points
4: **for** each epoch $e = 1, \ldots, E$ **do**
5:     **for** batch $b \in \mathcal{B}$ **do**
6:         $w \leftarrow w - \gamma \nabla f_i(w; b)$
7:         every $h$ batches pause training and return $w$ (resume at next call)
8:     **end for**
9: **end for**

---

# B  Additional Experimental Results

**Experimental Setup.** We use 16 agents to train a ResNet18 and a small convolutional neural network (for CIFAR-10 and MNIST respectively). The optimizer we use is Stochastic Gradient Descent for CIFAR-10 and Adam for MNIST. We use the standard FedAvg algorithm to perform federated training. We ran all experiments on a cluster of 2-4 GPUs, with the 16 CPUs (agents) pinned to a GPU. We use GeForce GTX 1080 Ti GPUs (11GB of memory) and the CPUs used are Xeon 4216. We provide a table of the hyperparameters not listed in Section 6 below.

Table 1: Hyper-parameters for CIFAR-10 Experiments.

| Model | Batch Size | Learning Rate | Training Cost | Epochs | Local FedAvg Steps $h$ |
|---|---|---|---|---|---|
| ResNet18 | 128 | 0.05 | 1.024e-07 | 100 | 6 |

Table 2: Hyper-parameters for MNIST Experiments.

| Model | Batch Size | Learning Rate | Training Cost | Epochs | Local FedAvg Steps $h$ |
|---|---|---|---|---|---|
| CNN | 128 | 1e-3 | 7.111e-08 | 100 | 6 |

**Test Loss and Accuracy Plots.** We include the test loss and accuracy plots, including error bars, below for federated and local training for CIFAR-10, MNIST, and HAM10000. We run each experiment three times. As expected, FL outperforms local training in the iid (left), mild non-iid (middle) and strong non-iid (right) settings. Federated training, via FedAvg, is more robust to non-iid distributions than local training. As detailed in Section 6, our non-iid distributions are Dirichlet with $\alpha = 0.3, 0.6$.

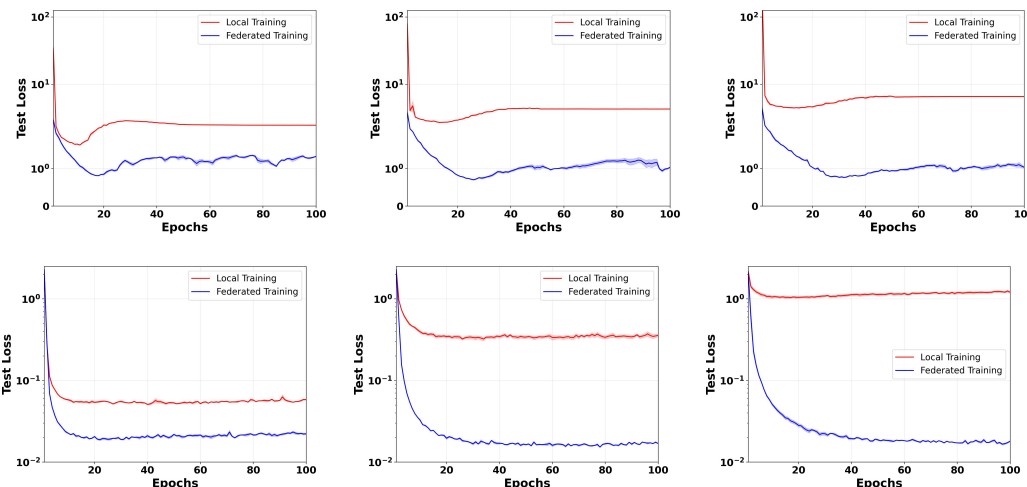

Figure 5: **Test Loss for CIFAR10 (top) and MNIST (bottom) in Heterogeneous Settings.** FL outperforms local training on iid (left) and mild (middle) & strong (right) non-iid Dirichlet settings.

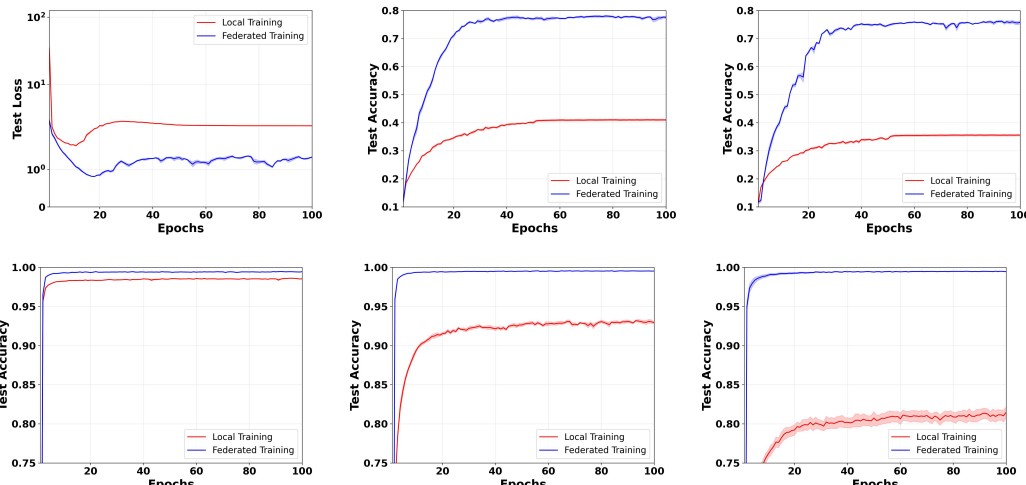

Figure 6: **Test Accuracy for CIFAR10 (top) and MNIST (bottom) in Heterogeneous Settings.** FL outperforms local training on iid (left) and mild (middle) & strong (right) non-iid Dirichlet settings.

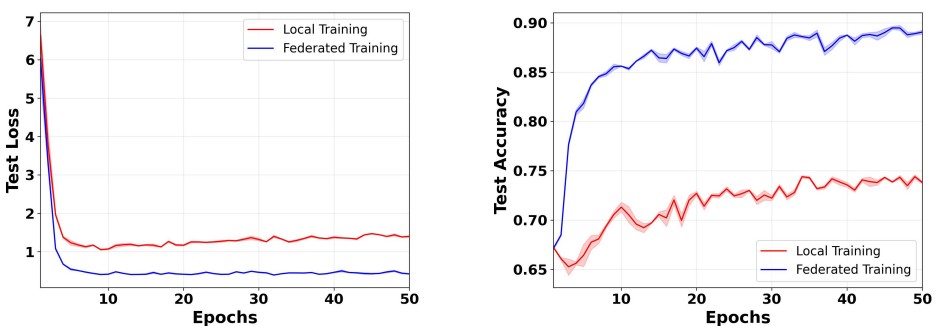

Figure 7: **HAM10000 Test Loss (Left) and Accuracy (Right) for Federated and Local Training.**

## C   Proofs

**Theorem 1** (Optimal Local Data Usage). *For an agent $i$ with marginal cost $c_i$, the optimal amount of data $m_{i,l}^*$ used for local training is $m_{i,l}^* := \sqrt{\frac{\gamma\sigma^2 L}{2c_i}}$.*

*Proof.* Each agent has a local loss function as shown in Equation (2). Taking the derivative of $\ell_{i,l}(m)$ with respect to the data contribution amount $m$ yields,

$$\frac{d\ell_{i,l}}{dm} = -\frac{\gamma\sigma^2 L}{2m^2} + c_i = 0 \quad \longrightarrow \quad m_{i,l}^* = \sqrt{\frac{\gamma\sigma^2 L}{2c_i}}. \tag{12}$$

$\square$

**Theorem 2** (Free-Riding: Optimal Federated Data Usage). *For an agent $i$ with marginal cost $c_i$, the optimal amount of data $m_{i,F}^*$ used for federated training is $m_{i,F}^* = \sqrt{\frac{\gamma\sigma^2 L}{2c_i}} - \sum \boldsymbol{m}_{-i}$.*

*Proof.* Each agent has a federated loss function as shown in Equation (3). Taking the derivative of $\ell_{i,F}(m)$ with respect to the data contribution amount $m$ yields,

$$\frac{d\ell_{i,l}}{dm} = -\frac{\gamma\sigma^2 L}{2(m + \sum \boldsymbol{m}_{-i})^2} + c_i = 0, \tag{13}$$

$$(m + \sum \boldsymbol{m}_{-i})^2 = \frac{\gamma\sigma^2 L}{2c_i} \quad \longrightarrow \quad m_{i,F}^* = \sqrt{\frac{\gamma\sigma^2 L}{2c_i}} - \sum \boldsymbol{m}_{-i}. \tag{14}$$

$\square$

**Theorem 3** (PFL Eliminates Free Riding). *For an agent $i$ with marginal cost $c_i$, the optimal amount of data $m_{i,PFL}^*$ used for federated training under Equation (4) is $m_{i,PFL}^* := \sqrt{\frac{\gamma\sigma^2 L}{2c_i}}$.*

*Proof.* Each agent has a penalized federated loss function as shown in Equation (5). Taking the derivative of $\ell_{i,PFL}(m)$ with respect to the data contribution amount $m$ yields,

$$\frac{d\ell_{i,PFL}}{dm} = -\frac{\gamma\sigma^2 L}{2(m + \sum \boldsymbol{m}_{-i})^2} + c_i + \frac{\gamma\sigma^2 L}{2(\sqrt{\frac{\gamma\sigma^2 L}{2c_i}} + \sum \boldsymbol{m}_{-i})^2} - c_i - 2\lambda_i\sqrt{\frac{\gamma\sigma^2 L}{2c_i}} + 2\lambda_i m = 0,$$

$$= 2\lambda_i\left(m - \sqrt{\frac{\gamma\sigma^2 L}{2c_i}}\right) - \frac{\gamma\sigma^2 L}{2(m + \sum \boldsymbol{m}_{-i})^2} + \frac{\gamma\sigma^2 L}{2(\sqrt{\frac{\gamma\sigma^2 L}{2c_i}} + \sum \boldsymbol{m}_{-i})^2} = 0. \tag{15}$$

Due to the convexity of Equation (5), as each piece of the utility function is convex, there is a single minimum which is carefully constructed to be at $m^* = \sqrt{\frac{\gamma\sigma^2 L}{2c_i}}$,

$$\frac{d\ell_{i,PFL}(m^*)}{dm} = 2\lambda_i\left(\sqrt{\frac{\gamma\sigma^2 L}{2c_i}} - \sqrt{\frac{\gamma\sigma^2 L}{2c_i}}\right) - \frac{\gamma\sigma^2 L}{2(\sqrt{\frac{\gamma\sigma^2 L}{2c_i}} + \sum \boldsymbol{m}_{-i})^2} + \frac{\gamma\sigma^2 L}{2(\sqrt{\frac{\gamma\sigma^2 L}{2c_i}} + \sum \boldsymbol{m}_{-i})^2},$$

$$= 0 \quad \longrightarrow m_{i,PFL}^* = \sqrt{\frac{\gamma\sigma^2 L}{2c_i}}. \tag{16}$$

$\square$

**Lemma 1** (Assurance of IR at Optimum). *Let $c_i$ be the marginal cost for an agent $i$, $m^* := \sqrt{\frac{\gamma\sigma^2 L}{2c_i}} = m_{i,l}^* = m_{i,PFL}^*$ be the agent's locally optimal data usage, and $\alpha \in [0,2)$ be a server-specified hyper-parameter. In order to ensure that the optimal penalized federated loss is at least lower than the optimal local loss, $\ell_{i,PFL}(m_i^*) \leq \ell_{i,l}(m_i^*)$, one must select $\lambda_i$ such that,*

$$\lambda_i := \frac{m^*(\sum \boldsymbol{m})}{(2-\alpha)\gamma\sigma^2 L \sum \boldsymbol{m}_{-i}}\left(c_i - \frac{\gamma\sigma^2 L}{2(\sum \boldsymbol{m}_{-i} + m^*)^2}\right)^2. \tag{17}$$

*Selection of such $\lambda_i$ results in a loss gap between $\ell_{i,PFL}(m^*)$ and $\ell_{i,l}(m^*)$ of,*

$$\Delta\ell_i := \ell_{i,l}(m^*) - \ell_{i,PFL}(m^*) = \frac{\alpha}{4}\left(\frac{\gamma\sigma^2 L \sum \boldsymbol{m}_{-i}}{m^*(\sum \boldsymbol{m}_{-i} + m^*)}\right). \tag{18}$$

*Proof.* We can determine the range of $\lambda_i$ values which ensure IR by plugging in $m_{i,l}^* = m_{i,PFL}^* = m^*$ into Equations (2) and (5) and finding the difference,

$$\ell_{i,l}(m^*) - \ell_{i,PFL}(m^*) = \frac{\gamma\sigma^2 L}{2}\left(\frac{1}{m^*} - \frac{1}{m^* + \sum \boldsymbol{m}_{-i}}\right) - \lambda_i\left(\frac{c_i}{2\lambda_i} - \frac{\gamma\sigma^2 L}{4\lambda_i(\sqrt{\frac{\gamma\sigma^2 L}{2c_i}} + \sum \boldsymbol{m}_{-i})^2}\right)^2,$$

$$= \frac{\gamma\sigma^2 L}{2}\frac{\sum \boldsymbol{m}_{-i}}{m^*(m^* + \sum \boldsymbol{m}_{-i})} - \frac{1}{4\lambda_i}\left(c_i - \frac{\gamma\sigma^2 L}{2(m^* + \sum \boldsymbol{m}_{-i})^2}\right)^2. \tag{19}$$

To ensure the Equation (19) is greater than or equal to 0, we must select $\lambda_i$ as the following,

$$\frac{\gamma\sigma^2 L}{2}\frac{\sum\boldsymbol{m}_{-i}}{m^*(m^* + \sum\boldsymbol{m}_{-i})} - \frac{1}{4\lambda_i}\left(c_i - \frac{\gamma\sigma^2 L}{2(m^* + \sum\boldsymbol{m}_{-i})^2}\right)^2 \leq 0,$$

$$\lambda_i \geq \frac{m^*(m^* + \sum\boldsymbol{m}_{-i})}{2\gamma\sigma^2 L\sum\boldsymbol{m}_{-i}}\left(c_i - \frac{\gamma\sigma^2 L}{2(m^* + \sum\boldsymbol{m}_{-i})^2}\right)^2. \tag{20}$$

Equation (20) shows that $\lambda_i$ must be non-negative (all values are positive and the squared term can be zero at a minimum). One can think of $\lambda_i$ as the parameter controlling the benefit received by an agent $i$. A larger value of $\lambda_i$ will result in a larger gap between $\ell_{i,l}(m^*)$ and $\ell_{i,PFL}(m^*)$ (Equation (19)). Inversely, a smaller value of $\lambda_i$ will result in a smaller gap between local and federated training utility. Therefore, we fully define $\lambda_i$ with a user or server specified hyperparameter $\alpha \in [0, 2)$,

$$\lambda_i := \frac{m_i(\sum\boldsymbol{m})}{(2-\alpha)\gamma\sigma^2 L\sum\boldsymbol{m}_{-i}}\left(c_i - \frac{\gamma\sigma^2 L}{2(\sum\boldsymbol{m})^2}\right)^2. \tag{21}$$

Plugging this term into Equation (19) yields the following utility gap between $\ell_{i,l}(m^*)$ and $\ell_{i,PFL}(m^*)$,

$$\Delta\ell_i := \ell_{i,l}(m^*) - \ell_{i,PFL}(m^*) = \frac{\alpha}{4}\left(\frac{\gamma\sigma^2 L\sum\boldsymbol{m}_{-i}}{m^*(\sum\boldsymbol{m_{-i}} + m^*)}\right). \tag{22}$$

$\square$

**Theorem 4** (Elimination of Federated Free-Riding With Truthful Agents). *PFL (Algorithm 1) using $\lambda_i$ from Lemma 1 is IR and eliminates the free-rider dilemma when agents are truthful.*

*Proof Reproduced from Section 4.* The result of Theorem 3 is that each agent $i$'s optimal strategy within the penalized federated scheme is to use their locally optimal amount of data $m_i^* = \sqrt{\frac{\gamma\sigma^2 L}{2c_i}}$. Furthermore, Lemma 1 states that using $m_i^*$ within the penalized federated scheme results in a reward, or improvement over local training, of $\Delta\ell_i = \frac{\alpha}{4}\left(\frac{\gamma\sigma^2 L\sum\boldsymbol{m}_{-i}}{m_i^*(m_i^* + \sum\boldsymbol{m}_{-i})}\right)$. Thus, by combining Theorem 3 and Lemma 1, truthful agents which choose to participate in PFL attain a reshaped and lower loss at the same optimum. Individually rational agents would therefore prefer to participate in PFL (Algorithm 1) over local training due to the reshaped optimum's lower loss. This optimum achieves the same data usage as local training, thereby eliminating the free-rider dilemma. $\square$

**Lemma 2.** *The probability of an agent "winning" in the truthfulness mechanism in Equation (10), given a reported cost $c$, is $\upsilon(c) := 2F_C(c)(1 - F_C(c))$, where $F_C$ is the CDF of $f_C$.*

*Proof.* The probability that an agent's cost $c$ is sandwiched in between two other randomly sampled agents' costs $c_u, c_v$ is equal to,

$$P(\text{sandwiched } c) = \upsilon(c) = P(c_u \leq c \leq c_v) + P(c_v \leq c \leq c_u). \tag{23}$$

Let $F_C$ be the cumulative distribution function (CDF) of the agent cost distribution $f_C$. The probability that a random cost, let's say $c_u$, is smaller than $c$ is equal to $F_C(c)$. The probability that a random cost, let's say $c_v$, is larger than $c$ is $1 - F_C(c)$. Since the costs are random, and the above situation could be flipped (*i.e.,* $c_v$ is smaller than $c$ with probability $F_C(c)$ and $c_u$ is larger than $c$ with probability $1 - F_C(c)$), the two probabilities in Equation (23) are equivalent. Thus, due to symmetry, $P(c_u \leq c \leq c_v) = P(c_v \leq c \leq c_u)$. Therefore, we can rewrite the equation above as,

$$\upsilon(c) = 2F_C(c)(1 - F_C(c)). \tag{24}$$

$\square$

**Theorem 5** (Main Theorem). *Each agent $i$'s best strategy, when participating in FACT (Equation 11), is to report its true cost and use its locally optimal amount of data $(m_i, c)_i^* = (\sqrt{\frac{\gamma\sigma^2 L}{2c_i}}, c_i)$.*

*Proof.* Agents participating in FACT follow the loss function shown in Equation (11),

$$\ell_{i,Fact}(m_i, c) = \frac{\gamma\sigma^2 L}{2(m_i + \sum \boldsymbol{m}_{-i})} + c_i m_i + P_{fr}(m_i, c) + P_{ct}(c). \tag{25}$$

Taking the expectation of the equation above over the distribution of agent costs $f_C$, and using the results of Lemma 2, yields,

$$\mathbb{E}_{f_C}[\ell_{i,Fact}(m_i, c)] = \frac{\gamma\sigma^2 L}{2(m_i + \sum \boldsymbol{m}_{-i})} + c_i m_i + P_{fr}(m_i, c) + \mathbb{E}_{f_C}[P_{ct}(c)],$$

$$= \frac{\gamma\sigma^2 L}{2(m_i + \sum \boldsymbol{m}_{-i})} + c_i m_i + P_{fr}(m_i, c) + \Delta\ell_i - \frac{3\upsilon(c)}{n} \sum_{j \neq i \in [n]} \Delta\ell_j \quad (26)$$

Taking the partial derivative with respect to $m_i$ and setting it equal to zero results in,

$$\frac{\partial}{\partial m_i} \mathbb{E}_{f_C}[\ell_{i,Fact}(m_i, c)] = \frac{\partial}{\partial m_i}\left[ \frac{\gamma\sigma^2 L}{2(m_i + \sum \boldsymbol{m}_{-i})} + c_i m_i + P_{fr}(m_i, c) \right] = 0 \tag{27}$$

From Theorem 3, we know that this results in $m_i^* = \sqrt{\frac{\gamma\sigma^2 L}{2c}}$. Now, when taking the partial derivative with respect to $c$ we can plug in $m_i^*$ (to ensure the critical point is evaluated as zero),

$$\frac{\partial}{\partial c}\left[ \underbrace{\frac{\gamma\sigma^2 L}{2(m_i^* + \sum \boldsymbol{m}_{-i})} + c_i m_i^* + P_{fr}(m_i^*, c) + \Delta\ell_i}_{\ell_{i,l}(m_i^*)} - \frac{3\upsilon(c)}{n} \sum_{j \neq i \in [n]} \Delta\ell_j \right] = 0. \tag{28}$$

By Lemma 1 and specifically Equation (7), we find that the first four terms above are equivalent to the local agent loss at its local optimum. Therefore, we can rewrite the equation above as,

$$\frac{\partial}{\partial c}\left[ \underbrace{\frac{\gamma\sigma^2 L}{2m_i^*} + c_i m_i^*}_{\ell_{i,l}(m_i^*)} - \frac{3\upsilon(c)}{n} \sum_{j \neq i \in [n]} \Delta\ell_j \right] = 0 \longrightarrow -\frac{3}{n} \sum_{j \neq i \in [n]} \Delta\ell_j \frac{\partial}{\partial c}\left[ \upsilon(c) \right] = 0,$$

$$-\frac{3}{n} \sum_{j \neq i \in [n]} \Delta\ell_j \frac{\partial}{\partial c}\left[ 2F_C(c)(1 - F_C(c)) \right] = 0. \tag{29}$$

The final line follows from Lemma 2. Since $F_C(c)$ is a CDF, its range is $[0,1]$. The function $\upsilon(c) = 2F_C(c)(1 - F_C(c))$ achieves is global maximum when $F_C(c) = 1/2$ (its critical point). Therefore, the value of $c$ which satisfies Equation (29) is one such that $F_C(c) = 1/2$. By Assumption 1, we know that the median value is assumed by each agent $i$ as $c_i$, since each agent believes its cost is equally likely to be larger or smaller than any other agent's cost. Thus, $F_C(c_i) = 1/2$ and $c^* = c_i$. Plugging this optimal value of $c$ back into the optimal value for $m_i$ leads to the following optimum: $(m_i, c)^* = (\sqrt{\frac{\gamma\sigma^2 L}{2c_i}}, c_i)$.

□

# D   Impact Statement & Limitations

Current federated systems are inhibited by agent free riding. The result is an unfair system. Some agents perform the bulk of training, while others sit idle. In the end, all the agents receive the same model performance. While there are current methods which eliminate agent free riding, they do not take into account agent truthfulness. Many agents can still provide false information to the server in order to free ride. The impact of FACT is that it provides an easily implementable mechanism which can make federated training more robust to free riding. Using FACT, agents are incentivized to no longer free ride even if they can lie about their training costs.

Our work faces limitations when agents and the central server act maliciously. On the agent side, we assume that agents are rational and do not collude with one another. Our future research direction is to prove that equilibria for FACT exist when agents are boundedly rational or colluding. On the server side, we assume that the central server acts honestly. In settings where the server cannot be trusted, new incentives or avenues must be built in order to ensure server honesty.

