# OpenReview forum: "FACT or Fiction: Can Truthful Mechanisms Eliminate Federated Free Riding?"
_NeurIPS.cc/2024/Conference — NeurIPS 2024 poster_

### Official Review · Reviewer_k1Mz · 2024-06-13

**Soundness:** 3
**Presentation:** 3
**Contribution:** 3
**Rating:** 6
**Confidence:** 3

**Summary:**

This paper introduces Federated Agent Cost Truthfulness (FACT), a novel mechanism that addresses the free-rider dilemma in federated learning. FACT ensures truthful reporting by agents and eliminates free-riding through a penalty system, competitive environment, and by encouraging agent participation with better performance than training alone. Empirically, FACT effectively avoids free-riding even when agents are untruthful and significantly reduces agent loss.

**Strengths:**

1. The method part of the paper has clear logic, and the contract theory is used to effectively prevent free-rider attacks in federated learning.
2. Almost all the proposed theories or definitions have been proven
3. The problem solved is very critical, and it is the first study of free-rider attacks in federated learning under active deception conditions (according to the author)

**Weaknesses:**

1. Eq 10, 26, and 27 lack punctuation.

2. Line 447, 452, 459, 475, 485, 496, and 514 are extra blank lines.

3. It may be better to add one figure in the Introduction to show the proposed issues.

4. The experimental part is not sufficient to thoroughly prove the effectiveness of the FACT mechanism. For example, it is obvious that the introduction of the FACT mechanism will bring additional communication overhead, but this part is completely missing in the experimental part.

5. The experimental part lacks organization. For example, I think the overall accuracy of the model prediction after using the FACT mechanism should be shown first, and then the various parts of FACT should be ablated, such as removing the Sandwich module, Penalized module, etc. Finally, some hyperparameters or other visualization experiments should be shown. At present, I am very confused about the results of the experimental part.

**Questions:**

1. The core contribution of the paper is to prevent free-rider attacks by participants in federated learning when they are able to cheat. And the author claimed FACT is the first truthful Horizontal FL method that eliminates free riding while simultaneously enforcing truthfulness. However, other federated learning algorithms also can prevent free-rider attacks, even though the authors claim that they cannot prevent participants from actively cheating. Wouldn't it be helpful to prove the effectiveness of the FACT if the paper add comparative experimental results of several related algorithms?

2. Is it a limitation that the parties cannot collude? If the parties collude, can FACT also effectively protect FL?

3. What is the additional computational overhead of FACT compared to traditional FL and local training?

4. According to Figure 2, why does using FACT to protect the FL process cause performance degradation? I noticed that the loss of FACT is higher than the loss of traditional FL.

5. I think the loss is not enough to reflect the performance of the model. When the model is overfitted, the loss increases but the accuracy still decreases. At the very least, the paper should show the accuracy of the final prediction effect (necessary) and add some ablation and hyperparameter experiments (after accept). I suggest you refer to these papers to show your experimental results if you still have the intermediate records of the experimental results.

[1] Fraboni, Yann, et al. Free-rider attacks on model aggregation in federated learning. International Conference on Artificial Intelligence and Statistics. PMLR, 2021.

[2] Lin, Jierui, et al. "Free-riders in federated learning: Attacks and defenses." arXiv preprint arXiv:1911.12560 (2019).

**Limitations:**

Based on the checklist, the author said "We provide the assumptions used for our work, as well as explain them, in Sections 3 and 5." However, I find it difficult to understand what the author wants to express in terms of specific limitations. I suggest writing it clearly at the end of the paper or in a separate section.

---

> ### Author Rebuttal · Authors · 2024-08-06
>
> Thank you, Reviewer k1Mz, for your insightful review of our paper. Below, we address all questions you raised.
>
> ## Addressing Weaknesses
>
> ---
>
> > **Weakness 1:** Eq 10, 26, and 27 lack punctuation.
>
> **Response to Weakness 1:** We begin by thanking you for catching the punctuation issues, and have fixed them within our paper.
>
> > **Weakness 2:** Line 447, 452, 459, 475, 485, 496, and 514 are extra blank lines.
>
> **Response to Weakness 2:** Each of these lines include an "end of proof" symbol on the far right of the line.
>
> > **Weakness 3:** It may be better to add one figure in the Introduction to show the proposed issues.
>
> **Response to Weakness 3:** We have added such a schematic into the Introduction of our paper.
>
> > **Weakness 4:** The experimental part is not sufficient to thoroughly prove the effectiveness of the FACT mechanism. For example, it is obvious that the introduction of the FACT mechanism will bring additional communication overhead, but this part is completely missing in the experimental part.
>
> **Response to Weakness 4:**
> - FACT does not incur extra communication costs during FL training.
> - FACT simply details a mechanism to distribute agent rewards after FL training is complete.
>
> FACT can be added to the end of any existing FL training method. The mechanism does not begin until after training finishes. As such, there is no added communication overhead during training.
>
> > **Weakness 5:** The experimental part lacks organization. For example, I think the overall accuracy of the model prediction after using the FACT mechanism should be shown first, and then the various parts of FACT should be ablated, such as removing the Sandwich module, Penalized module, etc. Finally, some hyperparameters or other visualization experiments should be shown. At present, I am very confused about the results of the experimental part.
>
> **Response to Weakness 5:** We thank the reviewer for their suggestions, and have included accuracy plots within our paper and will include ablations over our mechanism modules (Sandwich & Penalized) in our final version.
>
> - We have included the accuracy plots within our PDF rebuttal for the reviewer to view.
>
> ## Question Responses
>
> ---
>
> > **Question 1:** Wouldn't it be helpful to prove the effectiveness of the FACT if the paper add comparative experimental results of several related algorithms?
>
> **Response to Question 1:**
>
> - All related mechanisms would break down and allow free riding if agents could report untrue and inflated costs (detailed at the end of Section 4).
>
> We agree with the reviewer that in general, it would strengthen the work, but true to best of our knowledge ***our work is the first in the field*** to prove that a mechanism exists to eliminate FL free riding even when agents can be untruthful with the server.
>
> In summary, since we develop the first mechanism to eliminate free riding when agents can be untruthful, there are no comparable mechanisms to test against.
>
> > **Question 2:** Is it a limitation that the parties cannot collude? If the parties collude, can FACT also effectively protect FL?
>
> **Response to Question 2:** Collusion is indeed a limitation, one that is common to many truthful mechanisms (such as Vickrey auctions). We address our effort to more clearly detail and discuss our limitations [within our global rebuttal](https://openreview.net/forum?id=JiRGxrqHh0&noteId=w65DtMxzEP).
>
> > **Question 3:** What is the additional computational overhead of FACT compared to traditional FL and local training?
>
> **Response to Question 3:**  As detailed within our response above, FACT does not incur any additional computational overhead compared to traditional FL and local training during the actual training process.
>
> > **Question 4:** According to Figure 2, why does using FACT to protect the FL process cause performance degradation? I noticed that the loss of FACT is higher than the loss of traditional FL.
>
> **Response to Question 4:**
>
> - The loss of traditional FL is an upper bound: it's the loss that would occur if all agents did not free ride. Unfortunately, in practice, agents free ride and this loss will not be attained.
> - The degradation is a result of the penalties that are incurred in order to ensure agent truthfulness.
> - Even though FACT does not reach the upper bound, it is individually rational as it provides improved benefit to agents that choose to participate versus not.
>
> > **Question 5:** I suggest you refer to these papers to show your experimental results if you still have the intermediate records of the experimental results.
>
> **Response to Question 5:** As mentioned above, no comparable mechanism exists which can deal with agent untruthfulness.
>
> - Neither provided paper proposes a mechanism to defend against FL free riding through incentives, and neither deal with agent untruthfulness.
>
> The first paper deals with free-rider attacks (our paper is a defense) and the second focuses on defenses that utilize anomaly detection (we have cited both already within our paper).
>
> > **Question 6:** I think the loss is not enough to reflect the performance of the model. When the model is overfitted, the loss increases but the accuracy still decreases. At the very least, the paper should show the accuracy of the final prediction effect (necessary) and add some ablation and hyperparameter experiments (after accept).
>
> **Response to Question 6:** As mentioned above, we will include accuracy plots and ablation studies in our final version.
>
> We thank the reviewer for their great suggestions.  [We have included the accuracy plots in the global rebuttal as well](https://openreview.net/forum?id=JiRGxrqHh0&noteId=w65DtMxzEP).

---

> > ### Comment · Reviewer_k1Mz · 2024-08-08
> >
> > Thank you for your response. I have a few more questions:
> >
> > 1. RW3: I am curious about how much additional time FACT will add after training is completed. Although this is not part of the training phase, it still represents an additional time cost.
> >
> > 2. RW5: Do you mean you didn't have time to complete the ablation experiments? Why not present them in a table?
> >
> > 3. RQ1: I think the provided references can adopted as a baseline for comparison.
> >
> > 4. RQ4: Do you mean that if the parties do not free-ride, FACT will cause a decline in FL performance?

---

> > > ### Author Response · Authors · 2024-08-08
> > > **Response to Reviewer k1Mz**
> > >
> > > Thank you for your quick reply and continued discussion.
> > >
> > > ---
> > >
> > > > **Question 1:** I am curious about how much additional time FACT will add after training is completed. Although this is not part of the training phase, it still represents an additional time cost.
> > >
> > > **Response to Question 1:**
> > >
> > > The additional time that FACT adds after training is negligible. Here are the full steps that the server takes post-training:
> > >
> > > 1. Randomly group agents into threes.
> > > 2. Determine which agent has the middle reported cost within each group (any tiebreakers are randomly decided by the server).
> > > 3. Send rewards as prescribed in Equation (10) to the winning agent within each group.
> > >
> > > These three steps combined took no longer than five seconds within each of our experiments.
> > >
> > > > **Question 2:** Do you mean you didn't have time to complete the ablation experiments? Why not present them in a table?
> > >
> > > **Response to Question 2:**
> > >
> > > Yes. Due to running time-consuming additional experiments on real world datasets ([shown in our global rebuttal](https://openreview.net/forum?id=JiRGxrqHh0&noteId=w65DtMxzEP)) within a short time period, we were unable to complete the ablation experiments. We are committed to adding these ablation studies within our final version.
> > >
> > > > **Question 3:** I think the provided references can adopted as a baseline for comparison.
> > >
> > > **Response to Question 3:**
> > >
> > > The first provided reference, [1], is a paper detailing free rider attacks. Our paper proposes a novel mechanistic defense to disincentivize agents from free riding. Thus, [1] is not a baseline we can compare against.
> > >
> > > The second reference, [2], details both free riding attacks and defenses. The defenses proposed are not mechanisms, they are anomaly detection methods. As such, [2] does not disincentivize agents from free riding. The proposed methods simply allow a server to detect when free riding is occurring (thereby not solving the larger underlying issue that agents are *incentivized* to free ride). **FACT provably dissuades agents from free-riding, even when agents can be untruthful, eliminating the need to have an anomaly detection system.** Thus, [2] is not a baseline we can compare against as it is a detection method and not a mechanism.
> > >
> > > > **Question 4:** Do you mean that if the parties do not free-ride, FACT will cause a decline in FL performance?
> > >
> > > **Response to Question 4:**
> > >
> > > We believe there may be some confusion surrounding FACT's performance. Below, we take this opportunity to clarify the confusion.
> > >
> > > FACT does not cause a decline in FL model accuracy. The resultant model performs as well as the FL method utilized (since FACT does not alter training). However, in order to penalize and eliminate untruthfulness and free riding, FACT requires a contract fee from all participating agents. This fee is regarded as a loss for each agent, as shown in Equations (10) & (11). Fortunately, winning agents recoup a reward much larger than the contract fee (also shown in Equations (10) & (11)). Even with contract fees, agents provably achieve lower loss by participating in FACT than not (*i.e.,* FACT is Individually Rational).

---

> > > > ### Comment · Reviewer_k1Mz · 2024-08-08
> > > >
> > > > Thank you for your detailed explanation. My concerns almost have been solved. Although the experiment part of this paper has some limitation, the theory part is perfect. The logic is clear, there are no unknown symbols, and the relevant formulas have been proved accordingly. I think 6 is a proper score. I will raise my score.

---

> > > > > ### Author Response · Authors · 2024-08-08
> > > > >
> > > > > Thank you for your responsiveness and thoughtful discussion. We are happy to hear that you value our theory, and that our paper warrants acceptance!

---

### Official Review · Reviewer_hdyq · 2024-07-12

**Soundness:** 2
**Presentation:** 2
**Contribution:** 3
**Rating:** 6
**Confidence:** 2

**Summary:**

This paper introduces Federated Agent Cost Truthfulness (FACT) to tackle the issues of free riding and dishonesty in federated learning.  A penalty system is proposed to eliminate federated free riding. Meanwhile, a competitive environment is proposed to incentive agents provide truthful information.

**Strengths:**

1.	FACT addresses both the free riding and dishonesty issues in federated learning, making a highly significant contribution to the field.
2.	Sufficient theoretical evidence has been provided to demonstrate the effectiveness of FACT.
3.	This paper is well organized, ensuring it is easy to follow and offering a smooth reading experience.

**Weaknesses:**

1.	Are there any real-world examples that illustrate the necessity of addressing free-riding and dishonesty issues in federated learning?
2.	The paper states that "All rational agents seek to minimize their loss," but a more intuitive understanding is that rational agents seek to maximize their gains. There is a slight discrepancy between these two statements. How can this discrepancy be understood?
3.	Generally speaking, the payoff of a game is determined by the actions of all participants. Why, in Equation 2, can the payoff of agent i be calculated based solely on its own action mi ?
4.	Please elaborate more clearly on the logical relationship between Equation 2 and Equation 3.
5.	It would be better if the intuitive principles behind the formula design could be provided, such as for Equation 4.
6.	In Assumption 1, the assumption of "No Collusion" is made. Is this a simplified assumption, or is it also quite consistent with real-world federated learning scenarios?
7.	What are the limitations of the mechanism proposed in this paper?

**Questions:**

What if the individuals are boundedly rational?

---

> ### Author Rebuttal · Authors · 2024-08-06
>
> Thank you, Reviewer hdyq, for your insightful review of our paper. Below, we address all questions you raised.
>
> ## Addressing Weaknesses
>
> ---
>
> > **Weakness 1:** Are there any real-world examples that illustrate the necessity of addressing free-riding and dishonesty issues in federated learning?
>
> **Response to Weakness 1:** This is a good point. There are indeed lots of real world examples which illustrate the need of addressing free riding. We take this opportunity to expand upon this discussion as follows.
>
> - As detailed in [R1], free-riding has already been extensively studied and found in many peer-to-peer systems ([R2, R3]), such as BitTorrent ([R4]).
> - Realistic examples of FL free riding include training models for healthcare applications, [an example we provide additional experiments for in our global rebuttal](https://openreview.net/forum?id=JiRGxrqHh0&noteId=w65DtMxzEP).
>
> > **Weakness 2:** The paper states that "All rational agents seek to minimize their loss," but a more intuitive understanding is that rational agents seek to maximize their gains. There is a slight discrepancy between these two statements. How can this discrepancy be understood?
>
> **Response to Weakness 2:** Within literature, equilibriums can be found when either maximizing concave functions (gain) or minimizing convex functions (loss); there is an equivalence as they are the negative of one another.
>
> > **Weakness 3**: Generally speaking, the payoff of a game is determined by the actions of all participants. Why, in Equation 2, can the payoff of agent i be calculated based solely on its own action mi?
>
> **Response to Weakness 3:**
>
> - Equation (2) defines the *local* agent loss.
>
> The local agent loss describes the relationship between data quantity and loss for an agent training locally (by itself). Thus, other agent actions will not affect the agent's local loss.
>
> > **Weakness 4:** Please elaborate more clearly on the logical relationship between Equation 2 and Equation 3.
>
> **Response to Weakness 4:**
>
> - Equation (2) defines the local agent loss (training alone), while Equation (3) defines an agent's loss when it participates in distributed training.
>
> >  **Weakness 5:** It would be better if the intuitive principles behind the formula design could be provided, such as for Equation 4.
>
> **Response to Weakness 5:**
>
> - Equation (4) is designed such that first-order optimality conditions (zero gradient) of Equation (5) exists when an agent contributes its locally optimal amount data amount ($m_i^* = \frac{\gamma\sigma^2L}{2c_i}$).
>
> We thank the reviewer for this comment and agree that this should be provided within the paper. We have added this explanation within the paper.
>
> >  **Weakness 6:** In Assumption 1, the assumption of "No Collusion" is made. Is this a simplified assumption, or is it also quite consistent with real-world federated learning scenarios?
>
> **Response to Weakness 5:** Yes, the "No Collusion" assumption is standard within truthful mechanism theory (for example Vickrey auctions).
>
> - To the best of our knowledge, there hasn't been extensive evidence of collusion in FL. However, collusion is possible and this is a limitation of our work.
>  - Up until our work, it was an open question within FL whether a mechanism can provably guarantee agent truthfulness when querying their participation workload.
>  - We hope that our results drive further progress that can address new open questions that our work spurred such as: Can we find a solution when agents are boundedly rational or colluding?
>
> [R1] Lin, Jierui, et al. "Free-riders in federated learning: Attacks and defenses." arXiv preprint arXiv:1911.12560 (2019).
>
> [R2]  M. Feldman and J. Chuang, “Overcoming free-riding behavior in peer-to-peer systems,” SIGecom Exch., vol. 5, no. 4, pp. 41–50, Jul. 2005.
>
> [R3] M. Feldman, C. Papadimitriou, J. Chuang, and I. Stoica, “Free-riding and whitewashing in peer-to-peer systems,” IEEE Journal on Selected Areas in Communications, 2006.
>
> [R4] T. Locher, P. Moore, S. Schmid, and R. Wattenhofer, “Free riding in bittorrent is cheap,” in 5th Workshop on Hot Topics in Networks (HotNets), 2006.

---

> > ### Comment · Reviewer_hdyq · 2024-08-12
> >
> > Thanks for the author's responses. After reading them and other reviewer's comments, I would like to keep my score.

---

> > > ### Author Response · Authors · 2024-08-12
> > > **Reviewer hdyq Follow-Up**
> > >
> > > Thank you for your response! We wanted to follow-up to ensure that we have addressed all of your concerns. If not, we would be happy to continue discussion.

---

### Official Review · Reviewer_dpi3 · 2024-07-12

**Soundness:** 2
**Presentation:** 2
**Contribution:** 1
**Rating:** 5
**Confidence:** 5

**Summary:**

The paper "FACT or Fiction: Can Truthful Mechanisms Eliminate Federated Free Riding?" proposes a mechanism called FACT (Federated Agent Cost Truthfulness) to address the issue of free-riding in Federated Learning (FL). The key contributions include introducing a penalization scheme that incentivizes agents to provide truthful information, thereby eliminating free-riding and improving the performance of federated training. The proposed mechanism ensures that agents contribute as much data as they would in local training, thus aligning individual incentives with collective goals.

**Strengths:**

Originality: The paper introduces a novel combination of penalization and competition mechanisms to address free-riding and truthfulness in federated learning.

Quality: The empirical results demonstrate the effectiveness of the proposed mechanism in reducing agent loss and enforcing truthful behavior.

Significance: The problem of free-riding in federated learning is significant, and a robust solution would have considerable impact on the field.

Clarity: The explanation of the proposed mechanism is clear and detailed, making it easy to understand the core ideas.

**Weaknesses:**

Theoretical Reasoning: The paper does not explicitly define the game type and solution concept used, and lacks comparison with alternative abstractions. Recent critiques on the practicality of truthful mechanisms in complex environments are not addressed.

Practical Application: The focus on client-side truthfulness overlooks server-side challenges. Implementing the sandwich mechanism on the server side could increase costs and disincentivize its adoption. Clients might not trust the server to implement the mechanism, leading to untruthful behavior.

Experimental Validation: The experiments are limited to CIFAR-10 and MNIST datasets, and do not provide a broad validation of the mechanism's applicability and robustness across different scenarios.

**Questions:**

Can you explicitly define the type of game and solution concept used in the proposed mechanism? How does it compare with alternative abstractions in the literature?

How do you address recent critiques on the practicality of truthful mechanisms, that is abundant literature havs found that agents are not telling the true in truthful mechanisms, not even in the one with the strongest solution concept dominant strategies, especially in complex and distributed environments like federated learning?

What are the incentives for the server to implement the sandwich mechanism, given the potential increase in implementation costs? How do you ensure clients trust the server to follow through with the mechanism?

Can you provide more extensive experimental validation, possibly with different datasets and practical scenarios, to demonstrate the robustness and applicability of the proposed mechanism?

**Limitations:**

The paper acknowledges some limitations but does not sufficiently address the practical challenges and potential negative impacts of the proposed mechanism. Specifically:

Server-Side Incentives: The cost and incentive structure for the server to implement the mechanism is not thoroughly explored. This is crucial for practical adoption.

Broader Applicability: The experiments are limited in scope, and further validation with diverse datasets and scenarios is needed.

Trust and Adoption: The paper does not discuss how to build trust among clients that the server will implement the mechanism fairly, which is essential for the success of the proposed approach.

---

> ### Author Rebuttal · Authors · 2024-08-06
>
> Thank you, Reviewer dpi3, for your insightful review of our paper. Below, we address all the questions you raised.
>
> ## Question Responses
>
> ---
>
> > **Question 1:** Can you explicitly define the type of game and solution concept used in the proposed mechanism? How does it compare with alternative abstractions in the literature?
>
> **Response to Question 1:**
>
> - The type of game we consider is a One-Shot, Simultaneous Game.
> - The solution concept within FACT is a Pure Nash Equilibrium.
>
> Theorem 5 (Main Theorem) details FACT's Nash Equilibrium. Within the Proof of Theorem 5, we show that the best action for an agent, given the fixed actions of all other agents, is to contribute what is locally optimal and not to lie about its cost.
> - Alternative abstractions in FL Free-Riding mechanism have equivalent game types and solution concepts.
>
> The closest alternative abstractions within FL Free Riding, Blum et. al. (2021), Kamireddy et. al. (2022), Murhekar et. al. (2023), and Bornstein et. al. (2023), are also Simultaneous Move Games which all find Pure Nash Equilibriums. Therefore, our paper follows closely along with these related works. However, unlike these works, ***we are the first to deal with truthfulness in FL Free Riding***. Within each of the previous works, costs from each agent are publicly known and cannot be untruthful. We take a big step forward within the current literature by proposing the first truthful approach to eliminating FL Free Riding.
>
> We have added this discussion to our paper.
>
> > **Question 2:** How do you address recent critiques on the practicality of truthful mechanisms...?
>
> **Response to Question 2:**
>
> - The theory underpinning truthful mechanisms, like our own, is rigorous and will hold as long as the underlying assumptions are satisfied.
>
> Agents act untruthfully within provably truthful mechanisms only when the underlying assumptions of the mechanism are violated. Within FACT, we minimize the number of assumptions compared to related works and stick to only three assumptions standard within literature: agents (i) are rational and (ii) cannot collude with one another, and (iii) the server is honest in its mechanism implementation.
> - Theory provides important insight into the current limitations of truthfulness within the field and provides avenues for future progress into eliminating such limitations.
> - We are the first in the field to prove that a mechanism exists to eliminate FL free riding even when agents can be untruthful with the server.
>
> Although agents may act irrationally or collude in real-world scenarios, the insights that our theory provides is invaluable. Our theory allows researchers to better understand the nature of truthfulness within FL systems. Up until our work, it was an open question whether a mechanism can provably guarantee agent truthfulness when querying their participation workload. We hope to drive further progress that can address the new open questions that our work spurred: Can we find a solution when agents are boundedly rational or colluding? Can solutions be found when the server is dishonest?
>
> > **Question 3:** What are the incentives for the server to implement the sandwich mechanism...? How do you ensure clients trust the server...?
>
> **Response to Question 3:**
>
> - Eliminating free riding allows the central server to train higher-performing models.
>
> Without implementing the sandwich mechanism, devices will be untruthful and not use as much data during training (free-ride). Thus, the server is incentivized to implement the sandwich mechanism in order to eliminate the free-rider dilemma, increase the total amount of data used during training, and subsequently train a higher-performing model (as more data is used for training).
>
> - The server receives leftover agent contract fees.
>
> While the bulk of agent contract fees are returned to the winning agents after the sandwich mechanism, there are leftover fees that the server pockets. This is shown in Equation (10), where winning devices receive $\frac{3}{n}\sum_{j \neq i \in [n]} \Delta \ell_j$. The summation term is over just $n-1$ agents, while the outside factor divides by $n$. Thus, the server pockets the remaining winner contract fees (since no winning agent wins back its own fee) $\frac{3}{n}\sum_{k \in winners}\Delta \ell_k$.
>
> - Implementation costs in FACT are much smaller than related mechanisms.
>
> In the most comparable FL free-riding literature, Kamireddy and Bornstein, the server is required to send each agent a custom model with a prescribed accuracy. This means that the server has to perform additional training at the end of the normal FL training process. In FACT, the server only needs to match agents together, determine the mechanism winners, and disperse rewards to those winners.
>
> We have added this discussion to our paper.
>
> > **Question 4:** Can you provide more extensive experimental validation... to demonstrate the robustness and applicability of the proposed mechanism?
>
> **Response to Question 4:**
>
> - Our validation showcases that FACT performs well within multiple practical scenarios.
>
> Within our work we test the performance of FACT on two standard datasets, MNIST and CIFAR-10. We perform testing within IID and non-IID scenarios, both practical scenarios. In each of our experiments we find that an agent's best strategy is be truthful (Figure 1) and that agents indeed receive greater benefit by participating in FACT than by not participating (Figure 2).
> - Our validation is more extensive than those in related FL mechanism works.
>
> Comparable FL mechanism works either lack any validation on datasets (Kamireddy) or are limited to MNIST or EMNIST (Murhekar and Blum). We test on both MNIST and the larger and more difficult CIFAR-10 dataset.
> - **We have added additional validation in practical scenarios [within our global rebuttal](https://openreview.net/forum?id=JiRGxrqHh0&noteId=w65DtMxzEP) that demonstrate the effectiveness and robustness of FACT.**

---

> ### Author Response · Authors · 2024-08-12
> **Reviewer dpi3 Follow-Up**
>
> Thank you again for your insightful review! We wanted to follow-up to ensure that our rebuttal has indeed addressed all concerns listed within your original review. If not, we would be happy to continue discussion before the end of the discussion period tomorrow.

---

### Author Rebuttal · Authors · 2024-08-06

We would like to thank the reviewers for their constructive feedback. We address individual reviewer questions within their respective rebuttals. Below, we detail new experiments as well as a more thorough description of our paper's limitations.

## Real World Validation of FACT via Additional Experiments

---

We consider a realistic situation where a consortium of hospitals seek collaboration to train a model, privately in a FL manner, that can diagnose skin cancer. Now, let's say one of the hospitals is smaller and resource-constrained. It is difficult, but not impossible, for this hospital to collect more data for training. Thus, in the absence of a truthful FL mechanism, the smaller hospital could over-report its collection costs to the server in order to contribute little to no data towards training while still reaping the rewards of a well-trained global model. **We provide experiments that show FACT rebuffs this scenario, with figures detailing our results within the pdf rebuttal**.

To test this scenario, we train an image classification model on the HAM10000 [C1] dataset. HAM10000 consists of a collection of dermatoscopic images of pigmented lesions. The goal is to train a model which can perform automated diagnosis of pigmented skin lesions, including melanoma (skin cancer).

Our setup is similar to that within our original experiment section. One difference is that we fine-tune a ResNet50 model on HAM10000 that is pre-trained on ImageNet (a realistic approach for a hospital). HAM10000 is an imbalanced dataset, and evenly partitioning 80% of the data amongst 10 devices as local training sets further exacerbates the non-iidness of the data. This too is realistic, as data is often non-iid amongst FL agents. We use the Adam optimizer with a learning rate of 1e-3, batch size of 128.

- **FACT reduces agent loss by nearly 66% compared to local training.**
- **Agents maximize their reduction in loss compared to local training when they truthfully report their costs.**
- **Agents reduce their free-riding penalties when they use their locally optimal amount of data ($m^\star = 801$) for training.**

## Addressing the Limitations

---

We agree with the reviewers that a more detailed description and discussion of our limitations and impact is needed within our paper. Below, we expand upon FACT's limitations.

**(1) Agents are assumed to be rational and not collude with one another.**

Our future research direction is to prove that equilibriums for FACT exist when agents are boundedly rational or colluding.

**(2) The server is assumed to be honest.**

In settings where the server cannot be trusted, new incentives or avenues must be built in order to ensure server honesty.

[C1] Tschandl, Philipp, Cliff Rosendahl, and Harald Kittler. "The HAM10000 dataset, a large collection of multi-source dermatoscopic images of common pigmented skin lesions." Scientific data 5.1 (2018): 1-9.

---

### Decision · Program_Chairs · 2024-09-25

**Decision:**

Accept (poster)

**Comment:**

This paper introduces Federated Agent Cost Truthfulness (FACT) to address the issue of free-riding in federated learning. The key contributions include introducing a penalization scheme that incentivizes agents to provide truthful information, thereby eliminating free-riding and improving the performance of federated training. The proposed mechanism ensures that agents contribute as much data as they would in local training, thus aligning individual incentives with collective goals. Empirically, FACT effectively avoids free-riding even when agents are untruthful and significantly reduces agent loss. Some strengths of the paper include that: the method part of the paper has clear logic, and the contract theory is used to effectively prevent free-rider attacks in federated learning;
almost all the proposed theories or definitions have been proven, so sufficient theoretical evidence has been provided to demonstrate the effectiveness of FACT.; the problem solved is very critical, and it is the first study of free-rider attacks in federated learning under active deception conditions. In the rebuttal discussion, the authors had provided relatively concrete content to resolve many of reviewers’ concerns. There are still few concerns related to experiment parts in the discussion. After reading the reviews and the rebuttal, and going through the paper, my own reading is that the effectiveness of FACT mechanism has been demonstrated generally by the way of proving theories. Therefore, I suggest to accept this paper.